# Hyperthermic Intraperitoneal Chemotherapy for Primary or Recurrent Adrenocortical Carcinoma. A Single Center Study

**DOI:** 10.3390/cancers12040969

**Published:** 2020-04-14

**Authors:** Guido Alberto Massimo Tiberio, Vittorio Ferrari, Zeno Ballarini, Giovanni Casole, Marta Laganà, Michele Gritti, Elisa Arici, Salvatore Grisanti, Riccardo Nascimbeni, Sandra Sigala, Alfredo Berruti, Arianna Coniglio

**Affiliations:** 1Surgical Clinic, Department of Clinical and Experimental Sciences, University of Brescia at ASST Spedali Civili di Brescia, 25123 Brescia, Italy; zeno.ballarini@gmail.com (Z.B.); giovannicasole@gmail.com (G.C.); gritti.michele@hotmail.it (M.G.); elisa.arici90@gmail.com (E.A.); riccardo.nascimbeni@unibs.it (R.N.); arianna.coniglio@unibs.it (A.C.); 2Clinical Oncology, Department of Medical and Surgical Specialties, Radiological Sciences and public Health, University of Brescia at ASST Spedali Civili di Brescia, 25123 Brescia, Italy; drvittorio.ferrari@gmail.com (V.F.); martagana@gmail.com (M.L.); grisanti.salvatore@gmail.com (S.G.); alfredo.berruti@unibs.it (A.B.); 3Clinical Pharmacology, Department of Molecular and Translational Medicine, University of Brescia, 25123 Brescia, Italy; sandra.sigala@unibs.it

**Keywords:** adrenocortical carcinoma, hyperthermic intraperitoneal chemotherapy, tumor recurrence, safety, survival analysis

## Abstract

Background. This study explores the impact of Hypertermic Intra PEritoneal Chemotherapy (HIPEC) on adrenocortical carcinoma (ACC) management through a safety analysis completed by a preliminary evaluation of survival performances. Methods. Retrospective chart review of 27 patients submitted to surgical treatment completed by HIPEC for primary (SP, 13 patients) or recurrent (SR, 14 patients, 17 treatments) ACC. Safety was evaluated by means of procedural morbidity and mortality. Survival performances included multiple end points: local/peritoneal disease-free survival (l/pDFS), overall progression-free survival (OPFS), and overall survival (OS). Results. In the SP group, mortality was nil and morbidity was 46% (major 23%). At a median follow-up of 25 months, the median value for all the different survival measures had not been reached. Mortality was also nil in the SR group. However, morbidity was 77% (major 18%). Median l/pDFS and OPFS were 12 ± 4 and 8 ± 2 months, respectively. At a median follow-up of 30 months, median OS had not been reached. Conclusion. Surgery and HIPEC is an invasive procedure. Its employment in the surgery for primary setting deserves attention as it may affect oncologic outcomes positively. Its value in the management of recurrences seems less appreciable, albeit it may find its place in the multimodal management of a rare disease for which multiple therapeutic options do not yet exist.

## 1. Introduction

Surgical resection plays a pivotal role in the management of primary or recurrent adrenocortical carcinoma (ACC) [1]. Notwithstanding, disease control is difficult and recurrence rate is high. Abdominal recurrence was reported in up to 70–85% of cases [2,3,4] after resection of adrenal primary. Furthermore, in the setting of recurrent disease, it is known that curative redo resection offers the best therapeutic chances [1,5,6,7,8] yet the occurrence of “surgical cure” is anecdotal.

Cytoreductive surgery completed by Hypertermic Intra PEritoneal Chemotherapy (HIPEC) has been employed in the multimodal management of multiple primaries with unexpected survival performances in selected subgroups of patients affected by mesothelioma, pseudomixoma peritonei, colorectal, ovarian, and gastric cancer. These interesting results were achieved both in the prophylactic setting in patients without carcinomatosis but at high risk of developing peritoneal metastases and in the therapeutic setting in the management of advanced stage tumors involving the peritoneal surface [9].

This study investigates the impact of HIPEC in the management of ACC through a safety analysis and a preliminary evaluation of survival performances.

## 2. Patients and Methods

Data regarding patients submitted to surgical treatment and HIPEC for primary or recurrent ACC were reviewed retrospectively. Data were extrapolated from a prospectively collected institutional database and managed according to institutional rules, with patient consent. This research was approved by Comitato Etico di Brescia (Code NP 3430, permission date 21 March 2019).

From November 2006 to October 2019, 27 consecutive cases for which an abdominal curative resection was considered possible were treated. Indeed, exclusion criteria were basic: age over 75 (1 case); ECOG performance status > 2 (2 cases); and major lung, heart, or kidney deficiency (1 case). Patients with endovascular thrombus requiring major surgery on the cava vein with extracorporeal circulation (3 cases) were also excluded. 

Those patients submitted to regional adrenalectomy and HIPEC for primary ACC formed the “surgery for primary” (SP) group. In this case, HIPEC was employed in a prophylactic mode in order to reduce the local/peritoneal recurrence rate. The second study group, called “surgery for recurrence” (SR), included those patients submitted to cytoreductive surgery and HIPEC with the aim of treating recurrent disease. In this case, HIPEC played a therapeutic role.

All patients underwent thorough preoperative evaluation: medical history; physical examination; routine laboratory workup; cardiologic evaluation completed with echocardiography; and CT staging, MRI, and/or PET/CT scan whenever clinically indicated. Percutaneous biopsies of adrenal primaries were selectively employed.

Staging of primary ACC followed the ENS@T classification [10]. According to its anatomic localization, recurrence was described as loco-regional, when in the adrenal space or in its retroperitoneal surroundings; peritoneal, when in presence of peritoneal metastasis; and metastatic, which implies haematogenous diffusion to abdominal or extra-abdominal parenchyma.

The therapeutic strategy was discussed in a multidisciplinary adrenal session involving endocrinologists, oncologists, surgeons, and radiologists. As reported above, patients were submitted to this aggressive approach when a curative abdominal R0 resection was deemed feasible. Indeed, in selected cases, patients with extra-abdominal metastases responsive or stable after neoadjuvant chemotherapy were also submitted to this integrated treatment. 

### 2.1. Chemotherapy

Chemotherapy [1] was employed both in the neoadjuvant and the adjuvant setting. Standard schema was Etoposide, Doxorubicin, and Cisplatin associated to Mitotane (EDP-M). However, some variations occurred, taking into account previous oncological treatments. Adjuvant chemotherapy was indicated on the basis of pathological report and surgical outcome, while adjuvant mitotane was almost systematically employed. 

### 2.2. Surgical Technique

All patients in the SP group received a regional adrenalectomy in open surgery [11]. This consists of the removal of the adrenal gland en bloc with the renal capsule, of the lymph nodes at the renal hilum, and of all retroperitoneal soft tissues surrounding the cava vein, including the inter-aortic-cava space, which is dissected up to the aortic midline from the diaphragm down to the inferior mesenteric artery with the exposition of the right-hand surface of the coeliac trunk and superior and inferior mesenteric artery (right adrenalectomy). In the management of left-sided ACC, the midline limit of soft tissue clearance is the aortic midline, with exposition of the left-hand surface of visceral arteries; the clearance of inter-aorto-cava lymphatic tissues is limited to those surrounding the left renal vein. Appendix A summarize the procedures. Resection of surrounding organs such as the liver, kidney, spleen, or pancreas is only performed in the presence of direct infiltration; in the case of right-sided tumours, a sub-glissonian dissection is generally pursued in the quest for radicality.

Surgery for recurrent disease was more eclectic and adapted on a case-by-case basis. In general terms, a cytoreduction as is usually done for peritoneal diffusion of other primaries such as ovary, colonic, or gastric cancer was performed. Solid organ (liver, kidney, spleen, left pancreas, uterus, and annexes) and bowel sacrifices were performed as required, as was the extension of peritonectomy. When applicable, Peritoneal Cancer Index (PCI) was assessed at laparotomy; at the end of each procedure, the Completeness of Cytoreduction (CC) was defined. 

All surgery was performed by the same surgeon.

### 2.3. HIPEC

HIPEC was performed at the end of the surgical debulking using closed abdomen technique at 42 °C for 60 minutes with cisplatin (20 mg/m^2^/L) and doxorubicin (4.5 mg/m^2^/L) under constant extracorporeal manipulation of the patient’s abdomen to ease the uniform diffusion of perfusate. Perfusate was circulated at 0.6 L/min, approximately. Renal protection was obtained through venous infusion of sodium thyosolfate (loading dose 9 gr/m^2^ in 2 h followed by 12 gr/m^2^ during 12 h). Re-laparotomy with meticulous control of haemostasis and anastomoses concluded the surgical procedure.

### 2.4. Outcome

Safety evaluation was based upon procedural 90 days morbidity and mortality and upon description of perioperative complications which were also categorized according to Clavien-Dindo. For safety evaluations, all the procedures, including those performed in order to treat secondary recurrences, were considered.

Survival performances were calculated from the date of surgery and included multiple end points: local/peritoneal disease-free survival (l/pDFS), overall progression-free survival (OPFS), and overall survival (OS). All patients were followed by clinical examination and cross-sectional imaging (CT and/or MRI) performed every 4 months during the first 2 years after surgery, every 6 months until the 5th year, and annually thereafter. Disease recurrence was classified as described above. 

### 2.5. Statistics

Descriptive statistics are presented as median ± standard deviation and/or range. Survival measures were calculated from the date of resection to the date of censor or latest follow-up. Survival curves were generated by the Kaplan–Meier method.

## 3. Results

### 3.1. Surgery for Primary

In the SP group, 6/13 patients were male and median age was 54 (range 26–64). Their clinical characteristics are listed in Table 1.

In the 7 patients affected by stage IV ACC, metastatic lesions affected the lung in 5 cases and distant lymph nodes in 2 (mediastinal and subclavian nodes, respectively).

Preoperative biopsy of the tumor was achieved in 3 cases before neoadjuvant chemotherapy; in the remaining cases, a formal diagnosis was obtained through the association of imaging features and hormonal hypersecretion. 

Treatment details and immediate results are reported in Table 2. Neoadjuvant chemotherapy was standard (EDP-M) in 8 cases; 1 patient received EDP without mitotane, and 1 received mitotane alone. 

Postoperative course was spent at first (day 0) in the ICU and in surgical ward thereafter.

Ninety-day mortality was nil. The postoperative morbidity rate was 46%. In particular, 1 patient (8%) had a grade-4 complication: he developed an abdominal compartmental syndrome due to a postoperative haemorrhage of septic origin treated with re-laparotomy, haemostasis, and VAC-Therapy; he spent 96 days in the ICU and was discharged 7 days later.

Preoperative diagnosis of ACC was confirmed in all cases; Ki 67 > 10% was observed in 11 patients.

Postoperatively, 11 patients (84%) received mitotane. Within 6 months following surgery, 2 of those patients presenting with lung metastasis received a lung resection and the patient with mediastinal nodes received a mediastinal lymphectomy; thus, they reached the “non evidence of disease” status.

After a median follow-up of 25 months (6–168), 5 patients (38.5%) had a diagnosis of recurrence or progression: in 1 case (7.7%), it was peritoneal and haematogenous, and in the remaining 4, it was haematogenous: hepatic, lung, and/or distant nodes metastases in 3 patients and brain and bone metastases in 1.

At the time of the last follow-up, 2 patients died due to ACC recurrence and 1 died due to the progression of metastatic disease; 5 patients were living and disease free: 3 showed stable metastatic disease and 2 had ACC progression. Survival results are shown in Figure 1; for all the different survival measures, the median value had not yet been reached.

### 3.2. Surgery for Recurrence

The SR group enrolled 14 patients, 6 of whom were males, with a median age of 48 (range 30–72). Thirteen of them, at first treated elsewhere, had been addressed to our institution after detection of recurrence. 

In all cases, recurrence had been detected by follow-up cross-sectional imaging (CT or MRI). It followed open adrenalectomy in 2 cases and laparoscopic adrenalectomy in 5, while in 7 a laparoscopic procedure had been converted to open surgery due to difficulty in the dissection, to bleeding, and to rupture of the tumour. The median interval between evidence of recurrence and surgery was 13.5 months; in 9 cases, it was detected within 1 year after adrenalectomy (median 7 months) and it was detected in 5 > 1 year after adrenalectomy (median 25 months). At detection of recurrence, chemotherapy was started in 13/14 cases: EDP-M in 7 cases, mitotane alone in 5, and EDP in 1. One patient refused chemotherapy and asked for immediate surgery.

These 14 patients received a total of 17 cytoreductive procedures using HIPEC. Indeed, 3 of them received treatment a second time for a new recurrence, always detected within the first year after management of the first recurrence. At detection of second recurrence, the 3 patients who were reoperated on were already receiving adjuvant mitotane, yet in 1 case, EDP was associated. 

Patient and recurrence characteristics are resumed in Table 3.

Cytoreduction was CC0, CC1, and CC2 in 11, 1, and 2 cases, respectively. In the 3 patients submitted to redo procedures, a CC0 cytoreduction was always achieved. Besides peritonectomy, performed in those cases presenting peritoneal metastasis, associated resections were performed during all 17 procedures; in 2 cases, remnants of adrenal tissue from the first operation were also recognized and removed. Treatment details and immediate results are reported in Table 4. 

Postoperative course was spent at first (day 0) in the ICU and in the surgical ward thereafter.

Ninety-day mortality was nil. The postoperative morbidity rate was 77% (13/17). In particular, 2 patients were reoperated on for postoperative haemorrhage and 1, who developed a grade A pancreatic fistula, experienced a septic shock complicated by Addison and Guillan–Barré syndrome; he spent 35 days in the ICU.

Pathology confirmed preoperative diagnosis in 12 cases; Ki 67 > 10% was observed in 9 patients. In 2 patients (14.3%), a complete regression of the metastatic nodules after preoperative chemotherapy with post-apoptotic fibrotic tissue infiltrated by histiocytes (ypM0) was observed.

After a median follow-up of 30 months (6–58), a secondary recurrence was detected in 7/11 (63.6%) patients who benefitted from a CC0 resection. It was loco-regional and peritoneal in 3 cases, peritoneal and metastatic (abdominal parenchyma) in 2; and loco-regional, peritoneal, and metastatic in 2 cases. Recurrences were managed using repeated cytoreduction and HIPEC in 3 cases, using cytoreduction without HIPEC in 1 case and mitotane in 6 cases. Three patients received a 3rd cytoreduction without HIPEC and 2 of them received a 4th re-intervention. Abdominal disease progressed in all patients with CC1-2 resections despite mitotane and multiple chemotherapy schemas.

At the time of last follow-up, 4 patients died due to ACC progression, 6 were living with ACC relapse, and the remaining 4 were disease free. Survival results are exposed in Figure 2. Median l/pDFS was 12 ± 4.4 months, and median OPFS survival was 8 ± 2 months. Median OS had not yet been reached.

## 4. Discussion

This is the largest study evaluating the effect of HIPEC associated to surgical resection of primary or recurrent ACC. It evidences multiple points which deserve discussion, some of them of particular interest: the safety of this therapeutic strategy, the decision to implement the use of HIPEC in the prophylactic setting alongside the therapeutic setting, and finally the fact that HIPEC seems more effective in preventing than in treating recurrence.

### 4.1. Safety Considerations

In our experience, mortality rate was 0%. However, it was observed that the biological cost of the procedure was high. In the prophylactic and therapeutic subgroups of patients, overall morbidity was observed in 46% and 77% of cases, respectively, while major (grade 3 and 4) morbidity rates were 23% and 18%. As expected, overall morbidity was higher in the therapeutic setting due to the greater complexity of surgical procedures; to the fact that eclecticism plays an important role in this type of surgery which is less prone to an accurate preoperative planning; and to the higher rate of associated resections, performed during all procedures, with a median of 2.35 (1–6) removed organs per intervention. Notwithstanding, the rate of major complications was similar in the 2 subgroups of patients, suggesting the nonnegligible biological impact of this therapeutic strategy. Reoperation was always dictated by the need to control postoperative hemorrhage despite a systematic and punctilious control of any hemorrhagic source always being performed at the end of surgical demolition and repeated at the end of HIPEC. Pancreatic fistula was observed in all cases submitted to left pancreatectomy, but it was also observed in 22% of cases presenting a left adrenal cancer removed respecting the pancreas. The impact of postoperative complications may be devastating in those patients presenting adrenal suppression by mitotane, and cortical function must always be supported broadly. Special attention must be paid to the continuation of parenteral cortisone administration at least 2–3 days after restarting oral administration in order to prevent any possible bowel malfunction. In both cases with grade 4 complication, a relatively inadequate administration of cortisone was recognized at the very beginning of the complicating process, which had been located at resumption of oral intake. 

Altogether, the complication rate remained within the parameters which are normally accepted for major surgery completed using HIPEC [9,12,13], in any case higher than those observed after regional adrenalectomy alone: in our previous experience, the morbidity rate was approximatively 10% and the length of hospital stay was 8 days. An accurate case selection and the attention to the peculiarities of these particular patients may increase the overall tolerability of this aggressive therapeutic protocol.

### 4.2. HIPEC in the Prophylactic and Therapeutic Setting

HIPEC at completion of surgical resection of advanced stage abdominal primaries was proposed in order to improve loco-regional disease control. It exerts its toxic effect on residual neoplastic tissue through direct contact of highly concentrated chemotherapy while concurrent hyperthermia displays a synergistic therapeutic effect by enhancing drug’s pharmacological efficacy, by facilitating drug penetration through the peritoneal surface, and by promoting tumor cell apoptosis. Moreover, the recirculation of perfusate through the abdomen exerts a mechanical clearance of free neoplastic cells. In multiple tertiary centers, this technique is employed both in patients with peritoneal carcinomatosis and in those considered at high risk of postoperative peritoneal recurrence.

The extreme rarity of ACC explains the scarcity of literature exploring the role of HIPEC in the management of this particular neoplasm. However, for one case report [14] and 3 cases reported in a revision of multiple uncommon primaries [15], only 1 paper evaluated its prognostic impact on a series of 10 patients [16]. In all cases, however, HIPEC had been employed in the management of recurrent disease.

This is the first report regarding the use of HIPEC at the completion of regional adrenalectomy for ACC in order to improve the curativity of surgical procedure and to reduce the local and peritoneal recurrence rate. After curative resection of ACC, recurrence is more often local and/or peritoneal and generally occurs within the first 2 years after surgery. In our previous experience, local/peritoneal recurrence rate was approximatively 20% despite our aggressive surgical protocol. However, it had been reported as high as 75% [17,18]. When observed after adequately performed surgery, it is indicative of a non-emendable limit of the technique. One may speculate that it is consequence of seeding of viable ACC cells during the surgical procedure. These cells might originate from intraoperative tumor rupture yet also from blood or lymph spillage during dissection of masses which are correctly removed with no-touch technique. Considering the difficulties encountered in the surgical and medical management of recurrent ACC and the negative prognostic impact displayed by recurrence itself, we sought to optimize the quality of the surgical procedure. HIPEC appeared a promising tool for a number of reasons. At first, from the theoretical point of view, it potentially fulfilled all requisites capable of optimizing radicality. Furthermore, in our institution, it had already been implemented in the prophylactic mode in selected subgroups of patients with gastric and colon-rectal cancer.

On the contrary, the choice of HIPEC in the setting of local/peritoneal recurrent disease was more natural and immediate: it had already been described and the technique was employed in the setting for which it had originally been developed, with the aim of controlling microscopic residual after aggressive cytoreduction. 

### 4.3. Survival Performances

#### 4.3.1. Surgery for Primary

Our data suggest that HIPEC at the completion of regional adrenalectomy may be effective in preventing recurrence. This suggestion is strengthened by the fact that many of our patients presented negative prognostic factors associated to the likelihood of developing local/peritoneal recurrence: 10 patients (77%) had large and locally advanced or metastatic ACC, and in 11 cases, Ki-67 was > 10%. At the same time, however, it is important to signal the presence of possible confounding factors such as the large use of preoperative neoadjuvant chemotherapy and a sensible heterogeneity of clinical parameters. All the above considered, in our series, local/peritoneal recurrence rate was low, having been observed in only 1 case (7.7%) simultaneously with the dramatic progression of an already metastatic disease. At present, despite these positive preliminary data, too few elements to adequately evaluate the real impact of HIPEC in this particular setting are available. In fact, at a median follow-up of 25 month, the median value of each of the 3 considered survival measures had not yet been reached. 

Should this positive result be confirmed, it may have both direct and indirect positive impacts on clinical activity: direct by improving the radicality of surgical procedures and indirect by promoting the centralization of this rare disease into those major institutions which may display the complete “toolbox” for patient benefit. 

#### 4.3.2. Surgery for Recurrence

HIPEC also proved effective in pursuing local/peritoneal control of recurrent disease, albeit this result appears less appreciable than in the SP setting. It is difficult to compare our result with those reported in literature [17,18,19,20], as patient characteristics and treatment strategies may be different as are the considered end points of the studies. However, the median l/pDFS of 12 months observed in our cohort was almost double compared to those previously reported.

In reality, the only benchmark for comparison is the work by Hughes and Coll; they reported a longer peritoneal progression-free survival of 19 months [16]. Comparing this work to our experience, some differences were noted. A first point concerns patient selection. Our inclusion criteria had been liberal: HIPEC was employed in 87% (14/16) of recurrent patients operated on in the study period, and some of them were in suboptimal general status (ECOG 2). Rescue surgery for early recurrence (median 7 months) was performed in 9/14 cases, all patients presented multiple nodules of recurrence, and those patients for whom a CC0 cytoreduction could not be achieved were treated. All these clinical conditions are negative prognostic factors [8,19,20], and the exclusion of possible bad performers could have positively influenced survival outcome. A second point regards the technical aspect of HIPEC. In our study, the abdomen was perfuse with 2 agents at relatively low doses (cisplatin, 20 mg/m^2^/L and doxorubicin, 4.5 mg/m^2^/L) circulating for 60’ at 42 °C while Hughes and Coll used a single agent at high dose (Cisplatin, 250 mg/m^2^/L) circulating for 90’ at 40 °C. Both choices have theoretical advantages: synergistic chemotherapy effect and higher temperature on one hand and higher chemotherapy concentration, longer exposition, and longer mechanical clearance effect on the other. Since completeness of cytoreduction had been considered optimal (CC0) after 14/17 procedures, the fact that in our series HIPEC was less able to control microscopic tumor remnants must also be considered. It must be noted that our patients had been heavily pretreated with EDP (9/17) and that, thus, in 50% of cases, viable neoplastic elements could be resistant to chemotherapy. This argument has a rationale and is supported by the fact that the single patient who refused preoperative chemotherapy had the longest l/pDFS. One could evaluate the possibility to proceed to cytoreduction and HIPEC as soon as recurrence is detected, using a short time test in order to evaluate disease behavior. However, we consider that preoperative chemotherapy has at least 2 major advantages: the capability to effectively select the good candidate and the possibility to achieve the complete regression of the metastatic nodules (ypM0), as observed in 2 cases in our series.

From a different perspective, we also speculate that we had generally been unable to fully achieve the peritoneal clearance we thought we had attained, in particular, when recurrence followed laparoscopic surgery (12/14 cases in our series). This idea has some indirect support in literature [21]; it is generated by the fact that, in our experience, peritoneal recurrence appeared morphologically different after open or laparoscopic surgery. Following laparotomic surgery, peritoneal implants appear as typical tumors originating by a seeding element and suggest a discrete, relatively easy to control disease spread (Figure 3). It must be noted that, in the 2 cases observed after open surgery, l/pDFS had been of 14 and 21 months, respectively. After laparoscopic surgery and particularly, in the event of early recurrence and intraoperative rupture of the tumor, the peritoneum appears as if “sandblasted” by tumor cells (Figure 4). The positive intra-abdominal pressure exerted by CO_2_ pneumoperitoneum may play a decisive role in this phenomenon by stretching the mesothelial layer and thus facilitating tumor cell adhesion to the basal membrane. In these conditions, effectiveness of cytoreduction might be hampered by an excessive and potentially chemoresistant microscopic remnant.

In this particular subset of patients, however, the most important outcome measure is OS. From this point of view, our data are not so discouraging: at a median follow-up of 30 months, the median value had not yet been reached. This suggests that multimodal aggression of recurrent disease may positively affect survival.

### 4.4. Limits of the Study

The limitations of our work are clear and linked to its retrospective nature, to the limited number of patients, and to the relatively short follow-up. These limits, quite common when addressing such a rare disease, hamper our ability to evaluate the full clinical impact of the procedure. Furthermore, considering the SP group, one perceives the great variation of clinical stages which may act as a major confounder. However, in our case, the analysis of stage distribution, if anything, should bias the results unfavorably. To these biases, one must also add the impossibility of standardizing the surgical strategy in the subgroup of recurrent patients. One additional limit of our study is represented by the absence of a control group. This was motivated by excessive differences between the present and the historical experience which hindered an adequate case matching.

## 5. Conclusions

Our work reveals that surgery and HIPEC is an invasive procedure with a nonnegligible biological cost. Its employment in the surgery for primary setting deserves attention as it may effectively reduce local/peritoneal recurrences. Its value in the surgery for recurrence seems less appreciable, albeit it may find its place in the multimodal management of a rare disease for which we do not have multiple therapeutic options.

## Figures and Tables

**Figure 1 cancers-12-00969-f001:**
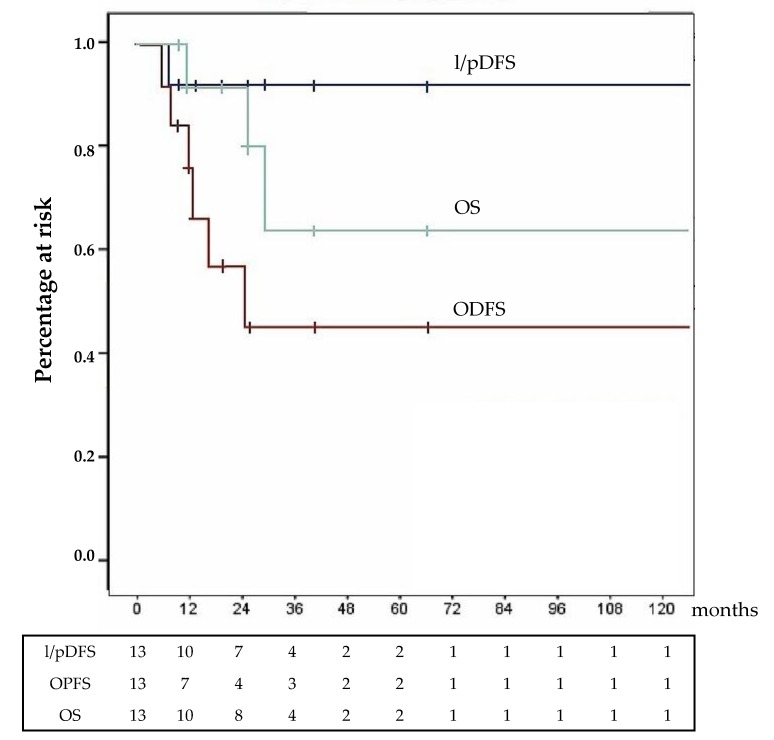
Survival analysis: Surgery for primary in 13 patients. l/pDFS = local/peritoneal disease-free survival; OPDS = overall progression free survival; OS = overall survival.

**Figure 2 cancers-12-00969-f002:**
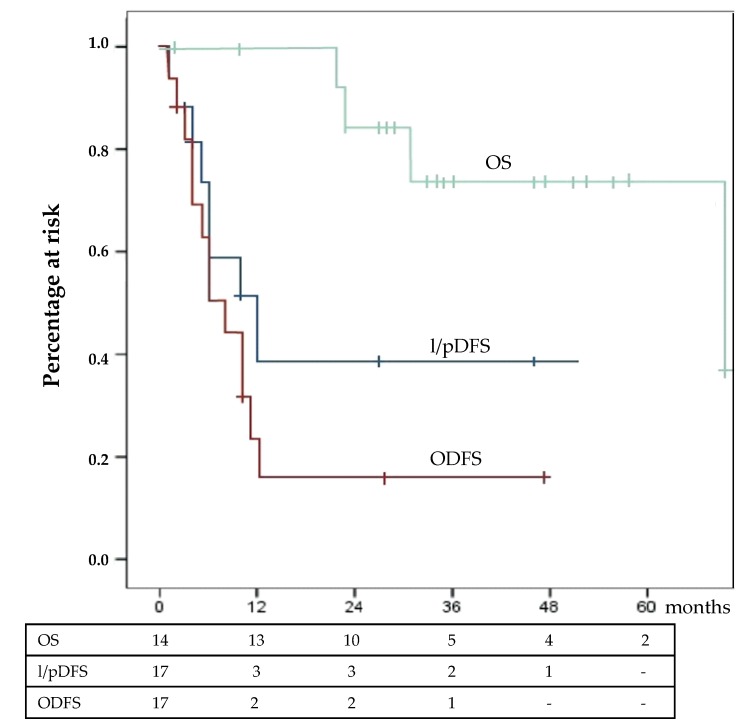
Survival analysis: Surgery for the recurrence group in 14 patients. l/pDFS = local/peritoneal disease-free survival; OPDS = overall progression free survival; OS = overall survival.

**Figure 3 cancers-12-00969-f003:**
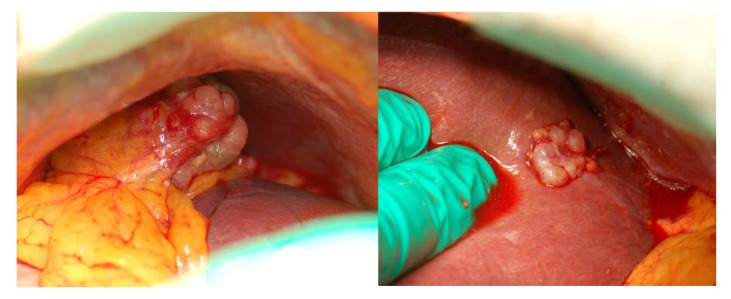
Neoplastic implant on the gastro-splenic ligament (**left**) and on the glissonian capsule (**right**) after open surgery.

**Figure 4 cancers-12-00969-f004:**
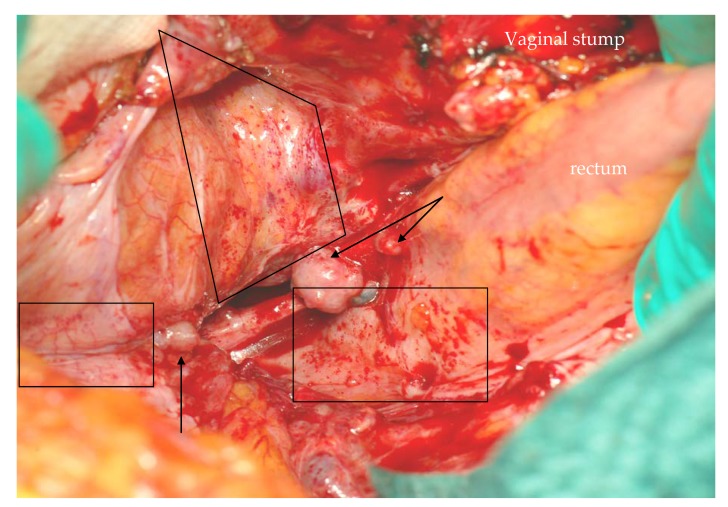
Carcinosis after laparoscopic surgery with tumor rupture: beside classic implants (arrows), the peritoneum appears sandblasted by tumor cells (tiny red spots in boxes).

**Table 1 cancers-12-00969-t001:** Surgery for primary: Clinical and tumor characteristics of 13 patients.

Variable	Measure
ECOG:	
0	10
1	2
2	1
ASA score:	8
2	
3	4
4	1
Right/Left adrenal tumor	2/11
Hyperfunction:	4
cortisol	
androgen	2
cortisol and androgen	1
Tumor diameter (cm, median ± SD)	10.5 ± 7.1
Ensat stage:	3
II	
III	3
IV	7
Positive regional nodes (pts)	3

**Table 2 cancers-12-00969-t002:** Surgery for primary: Treatment and immediate results for 13 patients.

Variable	Measure
Neoadjuvant CHT	9 (69%)
Duration of surgery	410’ (325’–630’)
Associated resections:	9 (69%)
Kidney/Spleen/Colon/Liver	7/2/1/1
Abdominal R0 resection	13 (100%)
Intraop. blood transfusion	5 (38%)
Units: median (range)	2 (2–4)
Postop. blood transfusion	6 (46%)
Units: median (range)	4 (2–54)
Postop. 90 days mortality	0 (0 %)
Postop. morbidity	6 (46%)
Grade 2:3 (23%)	Anemia: 3
	Pancreatic fistula: 1
Grade 3a: 2 (15%)	Pleural effusion: 2
	Pancreatic fistula: 1
Grade 4b: 1 (8%)	Septic shock Haemoperitoneum
Postop. relief of hyperfunction	7/7 (100%)
Hospital stay	14 days (7–109)
Adjuvant Mitotane	11/13 (84%)

**Table 3 cancers-12-00969-t003:** Surgery for recurrence: Patient and recurrence characteristics for 14 patients and 17 recurrences.

Variables	Measure
ECOG:	
0	7
1	5
2	2
ASA score:	
2	5
3	7
4	2
*Recurrence*	*n* = 14
Loco-regional	8
Loco-regional and peritoneal	3
Peritoneal	2
Peritoneal and hepatic	1
*II Recurrence*	*n* = 3
Peritoneal	2
Loco-regional and peritoneal	1
Peritoneal Cancer Index:	*n* = 9
median (range)	7 (2–9)

**Table 4 cancers-12-00969-t004:** Surgery for recurrence: Treatment and immediate results for 14 patients and 17 procedures.

Variable	Measure
Neoadjuvant CHT	16/17 (94%)
Duration of surgery (minutes)	445’ (288–600)
Associated resections	17/17 (100%)
Omentectomy:	9
Splenectomy:	7
Cholecystectomy:	5
Nephrectomy:	4
Hemicolectomy:	4
Left Pancreatectomy:	3
Diaphragm resection:	3
Appendectomy:	2
Oophorectomy:	2
Contra lateral Adrenalectomy:	2
Hepatectomy:	1
Intraop. blood transfusion	6/17 (35%)
Units: median (range)	2 (1–4)
R0 resection rate	14/17 (82%)
Postop. blood transfusion	9/17 (53%)
Units: median (range)	2 (2–4)
Postop. 90 days mortality	0/17 (0 %)
Postop. morbidity	13/17 (77%)
Grade 2: 10/17 (59%)	Severe anemia: 6
	Pleural effusion: 5
	Pneumonitis: 2
	Pancreatic fistula: 2
	Sepsis: 2
	Ileus: 1
Grade 3b: 2/17 (12%)	Haemoperitoneum: 2
Grade 4b: 1/17 (6%)	Pancreatic fistula, Septic shock Addison and Guillan–Barré
Hospital stay (17 procedures)	14 days (8–78)

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
