# Peer review of "Hyperthermic Intraperitoneal Chemotherapy for Primary or Recurrent Adrenocortical Carcinoma. A Single Center Study"

_cancers, 2020, doi:10.3390/cancers12040969_

Round 1

Reviewer 1 Report

The study has been well conducted and is to the point. The methods have been explained well. Results are clear and concise. I recommend it to be published.

Author Response

Dear Colleague, I really appreciate yuor kind judgement to our work

Reviewer 2 Report

Tiberio et al. reported a retrospective cohort that aimed to assess the safety and survival outcome of 27 patients who had cytoreduction for recurrent disease with HIPEC or primary adrenalectomy with adjuvant HIPEC. HIPEC protocol was 42C for 60 minutes with cisplantin and doxorubicin.  All patients with resection of primary tumors and HIPEC had R0 resection. Neoadjuvant EDP with or without M was used in 9 patients. Most patients with recurrent disease had EDP-M. Repeat cytoreduction and HIPEC were done in 3 cases. While there was no perioperative mortality, perioperative morbidity was high (77%), as expected, in those with recurrent disease and 46% in adjuvant HIPEC case. Recurrence detected by anatomic imaging studies was 64%. lpDFS and OPFS was 6 months in the abstract but in the manuscript stated l/pDFS was 12 month and OPFS was 8 months.

The manuscript is fairly well-written and is describing the 2 distinct cohorts. The interesting cohort is the adjuvant HIPEC that the authors boldly stated on page 11 lines 246-248 that it improves cure and reduce the local/peritoneal recurrence. The data on those with recurrent disease was, as expected, similar to the prior publication in ACC. The novelty may be in the adjuvant setting to shows that it was feasible but the benefit still remains to be studied. Thus, this manuscript essentially describes a single-center experience in HIPEC in primary and recurrent disease. Few comments are below.

  1. Please reconcile the lpDFS and OPFS.
  2. In the absence of control group and several confounding factors including EDP therapy, the claim mentioned above regarding adjuvant HIPEC on page 11, lines 246-248 is not supported by the data since the comparison is made with “generality” of disease behavior from authors’ impression. I would soften the statement considerably.
  3. Did the patient who had primary tumors resected plus adjuvant HIPEC have tissue diagnosis before proceeding with HIPEC? In other words, were there any patients who had HIPEC for other adrenal pathology that was thought to be ACC preoperatively?
  4. The concept of CO2 pneumoperitoneum causing carcinomatosis in ACC described on page 13 lines 320-323 is interesting and needs the reference.
  5. Page 5, postop morbidity: “n” number is needed in front of (15%) and (8%).
  6. The reviewer would want to know if HIPEC is more effective in locoregional control, compared to those without when patients are matched even retrospectively. Unfortunately, the manuscript does not have the controls.

Author Response

Please, see attachment

Reviewer 3 Report

The study from Tiberio et al. explores the impact of Hyperthermic Intraperitoneal Chemotherapy (HIPEC) in two subgroups of patients with adrenocortical cancer (ACC): 13 patients surgically treated for primary disease (SP) and 14 patients treated for recurrent disease (SR). ACC are rare and aggressive cancers for which progress in the treatment are required. HIPEC is an invasive procedure that can be helpful to improve oncological outcomes in some type of cancers, but its utility in the treatment of ACC has been scantly investigated and reported. Although median OS had not been reached at study termination, the study suggests a potential positive effect of HIPEC in the SP group of patents however it also shows a not negligible morbidity rate.

The presented study is interesting although it suffers of several limitation.    

Comments

  • The major limitation of the study is the heterogeneity of the evaluated population. The number of cases in each subgroup is small but still considerable for the rarity of disease, anyway the presence of a great variation in clinical parameters (i.e stage II; III and IV) among the considered patients might be a major confounder. Authors should comment this point and they should try to stratify patients at least for the main prognostic factors for ACC.  
  • A control group is missing, authors should at least comment on the expected oncological and safety outcomes of alternative treatment (i.e. surgery alone) in this type of patients.
  • Line 182 “Pathology confirmed preoperative diagnosis in 12 cases” authors should specify the pathology results in the remaining cases and add a comment on the possible effects of the HIPEC in these cases.

Reviewer 4 Report

The manuscript is of great importance to demonstrate the clinical results for hyperthermic IP chemotherapy for adrenocortical carcinoma. Hence I recommend to publish this present article as it's present form.

Author Response

Dear Colleague, thank-you very much for the king evaluation of our work

Round 2

Reviewer 2 Report

Thank you for your effort to address reviewer's comments. Most have been addressed as much as possible given the nature of the data.

Reviewer 3 Report

I do not have additional comments